# Remote Photoplethysmography with a High-Speed Camera Reveals Temporal and Amplitude Differences between Glabrous and Non-Glabrous Skin

**DOI:** 10.3390/s23020615

**Published:** 2023-01-05

**Authors:** Meiyun Cao, Timothy Burton, Gennadi Saiko, Alexandre Douplik

**Affiliations:** 1Department of Physics, Toronto Metropolitan University, Toronto, ON M5B 1E9, Canada; 2Department of Biomedical Engineering, Toronto Metropolitan University, Toronto, ON M5B 1E9, Canada; 3iBest, Keenan Research Centre of the LKS Knowledge Institute, St. Michael’s Hospital, Toronto Metropolitan University, Toronto, ON M5B 1E9, Canada

**Keywords:** microcirculation, photoplethysmography, PPG, remote PPG

## Abstract

Photoplethysmography (PPG) is a noninvasive optical technology with applications including vital sign extraction and patient monitoring. However, its current use is primarily limited to heart rate and oxygenation monitoring. This study aims to demonstrate the utility of PPG for physiological investigations. In particular, we sought to demonstrate the utility of simultaneous data acquisition from several regions of tissue using remote/contactless PPG (rPPG). Specifically, using a high-speed scientific-grade camera, we collected rPPG from the hands (palmar/dorsal) of 22 healthy volunteers. Data collected through the red and green channels of the RGB CMOS sensor were analyzed. We found a statistically significant difference in the amplitude of the glabrous skin signal over the non-glabrous skin signal (1.41 ± 0.85 in the red channel and 2.27 ± 0.88 in the green channel). In addition, we found a statistically significant lead of the red channel over the green channel, which is consistent between glabrous (17.13 ± 10.69 ms) and non-glabrous (19.31 ± 12.66 ms) skin. We also found a statistically significant lead time (32.69 ± 55.26 ms in the red channel and 40.56 ± 26.97 ms in the green channel) of the glabrous PPG signal over the non-glabrous, which cannot be explained by bilateral variability. These results demonstrate the utility of rPPG imaging as a tool for fundamental physiological studies and can be used to inform the development of PPG-based devices.

## 1. Introduction

Vital sign measurement is an invaluable medical methodology with many clinical applications ranging from routine clinical examinations to patient monitoring, and from the patient’s home to the bedside and to the operating room.

Photoplethysmography (PPG) is an optical modality for the non-invasive interrogation of tissue and is commonly used clinically for vital sign monitoring. The PPG signal itself is primarily low frequency in nature, as the main contributor is changing blood volume within the tissue [1]. PPG captures the pulse waveform, where the peak coincides with the arrival of the mechanical pulse wave caused by the left ventricular contraction. Despite its simplistic appearance, the PPG signal encodes a wealth of information. In addition to the heartbeat, which is the dominant contributor to the PPG signal, the PPG is modulated in both the amplitude domain and the frequency domain by multiple physiological processes, including respiration [2] and blood pressure oscillations (i.e., Mayer waves [3]). PPG has also been shown to be effective for capturing pathophysiology, such as atrial fibrillation [4].

PPG is typically acquired in contact with the skin, in one of two modes: transmission and reflection [1], with the former being most common clinically (i.e., pulse oximetry). PPG sensors designed in the transmission paradigm function by placing the emitter and detector in a position on the body such that light from the emitter can transit throughout the tissue and reach the detector. However, this is only possible in very specific places, such as the fingertips and earlobes [1]. Reflection mode PPG addresses this limitation by instead capturing back-scattered light, allowing the emitter and detector to exist in close physical proximity, therefore enabling PPG to be acquired from a wide range of body surfaces, given a sufficient size to sensor placement (forehead, arm, leg, etc.). Further, the tissue sampling volume can be regulated through design of the reflection mode PPG device through the manipulation of the distance between the emitter and detector [5]. Reflection mode is used for neuroscience research (functional near-infrared spectroscopy, or fNIRS [6]) and specialized clinical applications where it may be less practical to attach a transmission mode sensor, such as in neonatal patients [7]. Recently, reflection mode PPG has also found a niche in consumer health devices [8], as the transmission mode limitations do not constrain it.

Remote PPG, or rPPG, is an extension of the reflection mode PPG to a contactless mode. In this case, the typical sensor is a camera placed some distance from the patient. This modality has several substantial advantages over traditional (contact) PPG. Firstly, it does not require contact with the skin, which significantly simplifies clinical workflow, prevents potential cross-contamination, and is beneficial for patients with damaged or sensitive skin. Secondly, in contrast with the contact PPG, which is essentially a single-point measurement modality, rPPG allows the simultaneous collection of signals from large skin areas. Consequently, it will enable analyzing spatiotemporal characteristics of the PPG signal (e.g., pulse wave velocity, a measure of arterial stiffness [9]) and creating maps of tissue health indexes.

In particular, it can be helpful for the wider adoption of distributed clinical PPG measurements, such as the brachial-ankle pulse wave velocity (baPWV), which compares the timing of the arrival of the pulse wave at the brachial artery to the arrival at the ankle. baPWV has demonstrated utility in predicting major cardiovascular events [10]. However, the application of this technology is quite limited, which can be attributed, at least partially, to its current implementation that includes two detectors, which are placed on different body parts. In another example, rPPG has been shown in a research capacity to be capable of capturing the jugular venous pulse waveform [11,12], which is a valuable signal for diagnostic purposes.

In addition to clinical applications, the ability to collect signals simultaneously from large skin areas makes rPPG an invaluable investigational tool. However, it has not yet received proper attention.

Thus, a demonstration of the advantages of rPPG technology and its applicability for clinical research is of importance to increase awareness of rPPG. For example, its utility can be demonstrated by simultaneous data collection from different body parts, as it enables direct comparison between these signals, avoiding all complexities of synchronization of multiple independent single-point sensors.

One particular example is an investigation of differences between glabrous and non-glabrous skin. Glabrous skin composes approximately 10% of the skin on the body (appearing on the palms of the hands and soles of the feet, for example), and is distinct from non-glabrous (representing the remaining 90%, such as the backs of the hands and top of the feet, for example) through a lack of hair and melanin, as well as by increased thickness [13]. Glabrous skin also has distinctive microcirculation, with aterio-venous anastomoses transporting a significant volume of blood as compared with non-glabrous skin [14]. Considering these anatomical and functional differences in totality, we hypothesize that glabrous skin may generate significantly different PPG signals from non-glabrous skin.

We began to investigate this hypothesis in our previous work [15], where we used a consumer-grade camera (as embodied in a smartphone) to acquire rPPG signals at 60 frames per second (fps) to analyze the differences between glabrous (palmar) and non-glabrous (dorsal) skin in a pilot study. The video was acquired of both hands (one palm up and the other palm down) simultaneously since both were present in the frame, and the rPPG signals were isolated from each hand by segmenting the frame during post-processing. Across this pilot dataset, we discovered a statistically significant temporal lead time of the rPPG signal originating from glabrous skin as compared with non-glabrous (52 ± 36 ms) and a trend of an increase of glabrous rPPG amplitude as compared with non-glabrous (by 31%; however, it is not a statistically significant finding). While the use of a smartphone as the rPPG acquisition device demonstrated that it is possible to acquire the signal with a low-cost consumer camera, it was limited by frame rate, and thus signals with close temporal lags (such as between camera color channels within the same tissue) could not be observed.

Building upon our previous work, the current study aims to demonstrate the utility of rPPG by verifying these findings using: (a) a scientific-grade camera, (b) at higher acquisition speed with corresponded expanded set of analyses, and (c) on a larger pool of human subjects to derive statistically significant results. Thus, the novelty of the current work is that it is the first application of rPPG for the simultaneous data collection on two body parts (glabrous and non-glabrous skin in our case) that derives accurate and statistically significant results. As such, it reflects two significant study upgrading modifications. Firstly, in our previous work, we used a smartphone camera at 60 frames per second, while in our current work, we used a scientific-grade camera acquiring data at 500 frames per second. The increased frame rate enables the detection of more subtle temporal relationships and allows us to perform more in-depth analysis than was possible with the smartphone data, including temporal analysis between color channels. Secondly, given that the previous article described a pilot study, it contained a small number of subjects; in our current work, we expanded the number of subjects substantially to increase the rigor of our results.

## 2. Materials & Methods

### 2.1. rPPG System

The remote PPG was obtained with Mikrotron EoSens 1.1CXP2 camera (Mikrotron, Unterschleißheim, Germany), equipped with a Canon EF 50 mm f/1.8 STM lens (Canon, Tokyo, Japan), as shown in Figure 1. The camera had a total of 1280 × 864 active pixels, with an active sensors area of 17.50 × 11.80 mm and a corresponding pixel size of 13.7 × 13.7 μm, operating with a CMOS global shutter.

The camera was mounted on a tripod and directed downwards in an orthogonal manner toward the tissue, resulting in a distance from the lens to the tissue of approximately 80 cm. Two diffuse light sources (Neewer LED Video Light, Shenzhen, China) illuminated the tissue from around 45 degrees (90 degrees from each other) to minimize the presence of shadows on the tissue. The relationship between LED intensity (as measured by the Aurora4000 spectrometer; Changchun New Industries Optoelectronics Technology, Changchun, China) and the camera quantum efficiency (based on data from the manufactorer) is shown in Figure 2. The LED was supplied by mainline power at 60 Hz, which may cause variation in the spectrum, but that is above the frequency range of interest for this analysis.

Video data was captured at 500 fps (maximum jitter of ±4 ns and maximum frequency offset of 15 parts per million—neither of which can affect the results at 500 fps) at 512 × 512 pixels using the Bayer GR8 pixel format (enabling subsequent extraction of RGB pixel data). Four coaxial cables transmitted the acquired video data to the acquisition computer via the frame grabber. The acquisition computer ran the Linux operating system and was equipped with an Intel i5 processor and 32 GB of RAM. The frame grabber was a Euresys CoaXLink Quad CXP-12 (Euresys, Angleur, Belgium), connected to the acquisition computer via PCI. Python code on the acquisition computer interacted with the frame grabber via the Euresys eGrabber Python library to store the video data directly in the RAM. Once the entire video was collected, the video was written onto the disk, with an average file size of ~16 GB per video. The acquisition setup is shown graphically in Figure 3. The video was then post-processed, as will be described subsequently.

### 2.2. Experimental Protocol

The experimental protocol was performed as in our previous study [15]. Briefly, subjects rested for 10 min in the collection area before the initiation of data collection. The protocol then began with the subject in a seated position, with hands elevated over the head for 2 min. Next, the hands were lowered and placed on the video collection surface, with the right palm facing up (exposing the glabrous tissue) and the left palm facing down (exposing the non-glabrous tissue) (Figure 4). The video frame was configured to capture both hands simultaneously, with the hands centered in the frame to facilitate subsequent division into each hand during processing (Figure 4). Video was then recorded for 2 min. A 2-min rest period was taken post-recording, followed by a second 2-min hand elevation. The recording was then repeated with the hands in the opposite position (right palm down and left palm up). As previously described, the selection of hands as the source of the glabrous and non-glabrous tissue enforced an approximately equal distance to the heart, enabling temporal comparison between the two tissue types. Further, the requirement that each hand serve as both the source of the glabrous and non-glabrous tissue by performing a second recording for each subject enabled the assessment for bilateral differences, such as those that have been reported in blood pressure across the left and right arms [16], and those that may exist between dominant and non-dominant hands.

### 2.3. Subjects

Twenty-two healthy adult subjects participated in the study. The protocol and informed consent were approved through the Toronto Metropolitan University Research Ethics Board. Subjects included both males and females and in various skin tones. The increased study size as compared with our pilot project enables us to better assess the variation in our parameters, embedding normal physiological and anatomical variation across a healthy group of subjects. For instance, we expect that skin thickness varies across our group, which affects the resultant rPPG signal.

### 2.4. rPPG Extraction and Processing

All rPPG extraction and processing were performed in Matlab (Mathworks, Natick, MA) and in alignment with conventional rPPG methodologies [15]. Image frames were demosaiced to provide pixel intensity for each color channel (red, green, and blue), then a Gaussian filter (MATLAB imgaussfilt) was applied to reduce pixel noise. Regions of the frame corresponding to glabrous and non-glabrous tissue were manually identified. Each region’s pixel intensities for each color channel were averaged for each single frame (Equation (1), repeated for all color channels), resulting in an rPPG signal within each color channel at the same sampling frequency as the camera frame rate (i.e., 500 Hz). However, note that only the red and green color channels were used for analysis in this work, given that the blue channel has been shown to have the lowest SNR [17]. Specifically, in the blue range, the contribution of Raleigh scattering increases, which reduces the dynamic range of the channel.
(1)$$rPPG_{time=1}=\overset{pixel\;cols}{\underset{i=1}{\sum }}\overset{pixel\;rows}{\underset{j=1}{\sum }}pixel\;intensity_{frame=1,i,j}$$

Wavelet filtering was performed to reduce the signal to the frequency range of 0.5–2.5 Hz (MATLAB cwt & icwt), intended to focus on the cardiac contribution to the PPG signal (excluding lower-frequency signals from respiration, for instance). The first five and last five seconds were removed from the signal to exclude transients. The wavelet coherence (MATLAB wcoherence) was then computed between the glabrous and non-glabrous signals from each color channel for the purpose of signal quality assessment. The signals are expected to be minimally coherent, given that both are measuring a common physiological source and therefore only a constant relative phase difference should emerge from different measurement locations. If signals are not minimally coherent, then it is likely that one or both of the signals are contaminated with noise and not suitable for analysis. The temporal lag was then analyzed using cross-correlation (MATLAB xcorr) between the glabrous and non-glabrous signals within each color channel in temporal regions with the coherence of 0.33 and greater (on a scale of 0–1) and with signal amplitude summed across both glabrous and non-glabrous not exceeding five standard deviations. The temporal lag analysis was also applied to the red channel and green channel within the glabrous skin and the red channel and green channel within the non-glabrous skin, with a minimal coherence of 0.33 acting as the requirement for temporal inclusion. Next, the ratio of the amplitudes of the glabrous signal compared with the non-glabrous signal was calculated per each color channel using the envelope methodology previously described in [18], with values greater than 1 indicating a higher amplitude in glabrous. The signal processing methodology described here was applied to all acquired signals. The complete processing flow is shown in Figure 5.

### 2.5. Statistical Analysis

A threshold on signal quality was established to require that at least 80% of it (by time) meet the coherence (0.33) and amplitude (five standard deviations) levels in the green channel, since it has maximal sensitivity to hemoglobin changes across the three color channels. Any signal not meeting the signal quality threshold was excluded from the analysis. Statistical testing was performed on the parameters described above extracted from the remaining signals. Specifically, one-sided *t*-tests with α = 0.05 were performed to determine if the amplitude ratios of each color channel exhibited were greater than one. Further, one-sided *t*-tests with α = 0.05 were performed to determine if the lags between the glabrous and non-glabrous skin (from each color channel) were significant, expected those to be negative by the order supplied to the cross-correlation analysis, indicating that the glabrous skin signal is leading that of the non-glabrous. The same statistical test was applied to the red and green channels within the glabrous skin and the red and green channels within the non-glabrous skin.

Further, collecting signals with hands in each position (i.e., with the left hand acting as both the source of the glabrous skin and non-glabrous skin and the same from the right hand for each subject) enabled bilateral testing to determine if differences exist between these conditions. Specifically, the paired-sample *t*-test was used to determine if the differences between the paired observations originate from a distribution with a mean of zero or a non-zero mean.

## 3. Results

An exemplar pair of rPPG signals, simultaneously acquired from glabrous and non-glabrous skin, is plotted in Figure 6. The figure shows both the complete signal and the first 10 s, to provide greater resolution on the amplitude and timing of the two signal sources. Visually, in this example, greater amplitude is apparent in the glabrous signal as compared with that of the non-glabrous, and the peaks in the glabrous signal occur before those in the non-glabrous (indicating a glabrous lead over non-glabrous).

As described above, signals from 22 subjects were acquired. Post signal quality, 18 subjects remained with a total of 32 signals. The results of the statistical testing described are shown in Table 1, with all findings statistically significant at *p* < 0.05. Specifically, the amplitude ratio of glabrous over non-glabrous was found to be 1.41 ± 0.85 in the red channel and a higher 2.27 ± 0.88 in the green channel. Further, the temporal lead of glabrous over non-glabrous was 32.69 ± 55.26 ms in the red channel and 40.56 ± 26.97 in the green channel, and the distribution across the dataset is shown in Figure 7. Finally, within the glabrous skin, the red channel demonstrated a temporal lead over the green channel of 17.13 ± 10.69 ms and within the non-glabrous skin of 19.31 ± 12.66 ms.

Paired testing was performed as described above for the 14 subjects for which complete pairs were available. Regarding the parameters presented in Table 1, the only significant differences found were in the amplitude ratio in the red channel and the lead of glabrous over non-glabrous in the red channel.

The amplitude ratio in the red channel was 1.84 ± 0.81 in the left hand (i.e., when the left hand was used as the source of the glabrous skin) and 1.09 ± 0.79 in the right hand. The lead of glabrous over non-glabrous in the red channel was −19.71 ± 46.96 ms in the left hand and −61.29 ± 30.85 in the right hand. These results can be found in Table 2.

## 4. Discussion

Our study demonstrates the feasibility of rPPG as a tool to investigate tissue physiology. In particular, it can be used to extract and compare data from large skin areas or several skin areas without the need to synchronize data collection.

Our data shows several important findings.

Firstly, we obtained statistically significant results to demonstrate that the PPG signal in glabrous skin is stronger than in non-glabrous skin. Prior to the current study, these results were derived on a single subject [19] or a small number of subjects [15], which did not allow any feasible extrapolation.

Secondly, our data exhibited a statistically significant lead time of the glabrous signal over the non-glabrous (32.69 ± 55.26 ms in the red channel and 40.56 ± 26.97 ms in the green channel). Furthermore, we were unable to detect a bilateral effect that may account for both of these differences. These data support our previous findings using smartphone cameras on a small number of volunteers (52 ms in the green channel [15]).

Thirdly, we found a statistically significant lag in rPPG signals measured by red and green channels. This lag can be attributed to the deeper penetration of light in the red range of the spectrum. Thus, the red light scoops the signal from deeper layers (e.g., deep plexus), which experience pulse wave propagation earlier than the superficial plexus and capillaries. This result is in line with results reported by Moco [19] using contact PPG. Furthermore, the results are consistent for glabrous and non-glabrous skin, which can be considered as an indirect confirmation of essentially similar PPG mechanisms in both skin types. Importantly, this finding (17.13 ms and 19.31 ms between the color channels in the glabrous skin and non-glabrous skin, respectively) was not possible without the use of the high-speed camera, which at a frame rate of 500 fps, records a frame every 2 ms. In contrast, our previous work used a consumer smartphone, which captured data at 60 fps, and therefore a frame every ~17 ms.

In addition, we found an interesting dissimilarity between the left and right hands. In particular, there is a statistically significant difference in rPPG amplitude between glabrous and non-glabrous skin and lead of glabrous over non-glabrous when using the left hand as the source of the glabrous tissue as compared with the right hand as the source, both only found in the red channel (see Table 2). While the nature of this asymmetry is unclear, it can potentially be attributed to the concept of the dominant hand. It is known that 90% of the population is right-handed. As we have not traced which hand of the participant is dominant, this effect can likely be attributed not to the left vs. right but rather to the dominant vs. non-dominant hand. In particular, we can speculate that in comparison to the non-dominant hand, the skin on the palm of the dominant hand may have a thicker epithelium layer (reducing PPG amplitude) and perhaps a more developed microvasculature (increasing PPG amplitude). These findings warrant further investigation and will be explored in future work.

While we found interesting differences between glabrous and non-glabrous skin, there is much left to learn. For instance, a high proportion of flow in glabrous skin occurs through arterio-venous anastomoses, or AVAs, which may contribute to the observed differences. However, this hypothesis is not supported by the finding that the time lag between red and green signals is essentially identical in glabrous and non-glabrous skin. Thus, this hypothesis will require further exploration. Similarly, given that AVAs are a critical thermoregulatory mechanism in the thermoneutral zone, the lag may be affected by ambient temperature in the subject’s environment. It can be explored in further work. Further, AVA activation is oscillatory [19] with several cycles per minute. Perhaps, exploring lower frequencies (on a scale of 0.05–0.1 Hz) is required, which can be a focus of future research.

The presented findings can have several practical applications. In particular, some PPG-based technologies may be directly affected by the temporal lag between the glabrous and non-glabrous skin of the hand. For instance, pulse wave velocity [9] could be measured using PPG sensors placed at multiple locations on the body, with applications including baPWV [10]. Suppose the sensors are placed on both glabrous and non-glabrous skin. In that case, the technology developers should be cautious of the effect of the lead time of glabrous skin over non-glabrous when calculating the pulse wave velocity. In addition, with the wide adoption of wearable devices exploring unconventional PPG acquisition sites, our findings can be used to optimize the location and increase signal-to-noise ratio (SNR) as explained below.

In future work, we plan to expand the analysis to a broader frequency range and correlate results with a clinical gold standard to understand the origins of such findings. Additionally, the optimization of the acquisition setup was out of the scope of the current work, which we plan to explore in future work, with the goal of increasing signal to noise ratio. For example, customized filtering may be applied to optimize light source emission in combination with camera quantum efficiency according to the skin absorption spectrum. Specifically, narrow band imaging has been shown to improve signal to noise ratio [20]. Further, the optimization of the acquisition setup should consider that all work to date has taken place in a laboratory environment with ideal conditions, and that data collection in future applications may take place under non-ideal conditions. Therefore, the sensitivity of the sensors to environmental conditions must be quantified and managed through design improvements of the system, as necessary. Finally, we are not currently proposing clinical applications, such as disease screening or diagnosis, but are rather aiming to demonstrate the broad utility of rPPG through its use as a tool for physiological investigations. However, we may target clinically-oriented applications in the future.

## 5. Conclusions

Photoplethysmography is a noninvasive optical technology with applications including vital sign extraction and patient monitoring. However, its current use is limited and confined mainly to heart rate and oxygenation monitoring. This study aimed to broaden PPG utility by demonstrating the value of PPG for physiological investigations. As such, we presented the proof-of-concept for using rPPG for the simultaneous data collection on two body parts (glabrous and non-glabrous skin of hands in our case) that derive accurate and statistically significant results. Our results support the utility of large-area mapping remote PPG measurements. In particular, it can be used to extract and compare data from large skin areas or several skin areas without the necessity of synchronizing data collection.

Our results confirmed previous findings obtained by the smartphone camera. It allows extending rPPG applications to consumer-grade cameras, which can be used in multiple care settings, including patient homes. Bringing diagnostics closer to the patient improves health outcomes and reduces healthcare costs.

As a result of the increased frame rate and a larger pool of subjects, we were able to derive accurate and statistically significant results. In particular, we found a statistically significant difference in the amplitude of the glabrous signal over the non-glabrous signal (1.41 ± 0.85 in the red channel and 2.27 ± 0.88 in the green channel). In addition, we found a statistically significant lead of the red channel over the green channel. The results are consistent between glabrous (17.13 ± 10.69 ms) and non-glabrous (19.31 ± 12.66 ms) skin. We also found a statistically significant lead time (32.69 ± 55.26 ms in the red channel and 40.56 ± 26.97 ms in the green channel) of the glabrous PPG signal over the non-glabrous, which cannot be explained only by bilateral variability.

In addition, we found an interesting difference between the left and right hands in rPPG amplitude between glabrous and non-glabrous skin and lead of glabrous over non-glabrous, both observed in the red channel. We speculate that this asymmetry can be attributed to the dominant hand. However, it requires further investigation.

These results can be used in developing and optimizing PPG-based devices and methodologies.

## Figures and Tables

**Figure 1 sensors-23-00615-f001:**
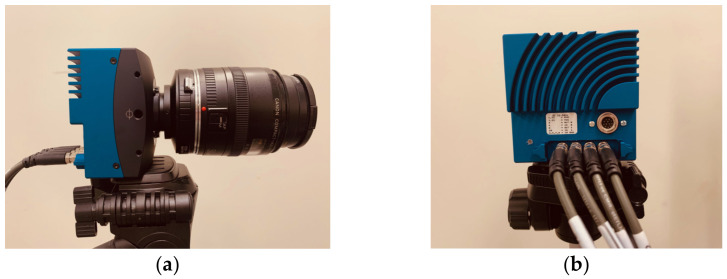
Mikrotron EoSens 1.1CXP2 camera used for data acquisition, with side (**a**) and back (**b**) views.

**Figure 2 sensors-23-00615-f002:**
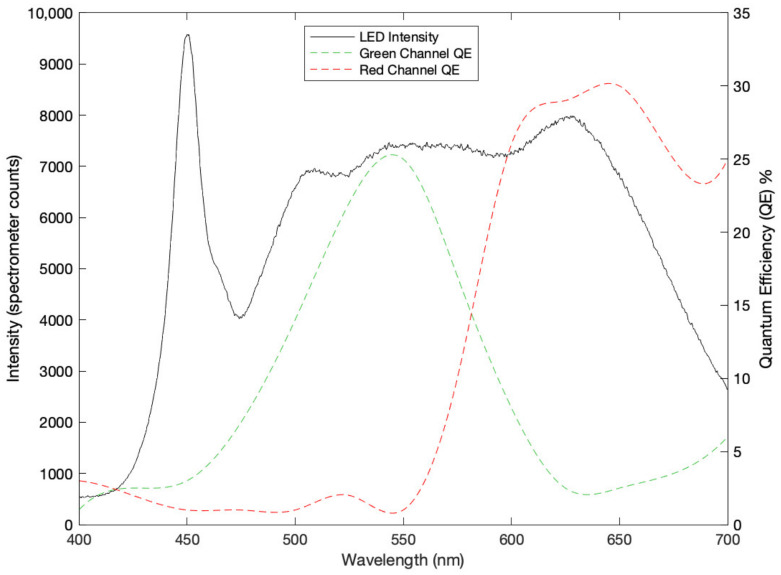
Relationship between the LED intensity (black line, with axis on the left), as measured by spectrometer counts, and quantum efficiency of the camera’s red and green channels (plotted in their own colors, with axis on the right), from 400–700 nm.

**Figure 3 sensors-23-00615-f003:**
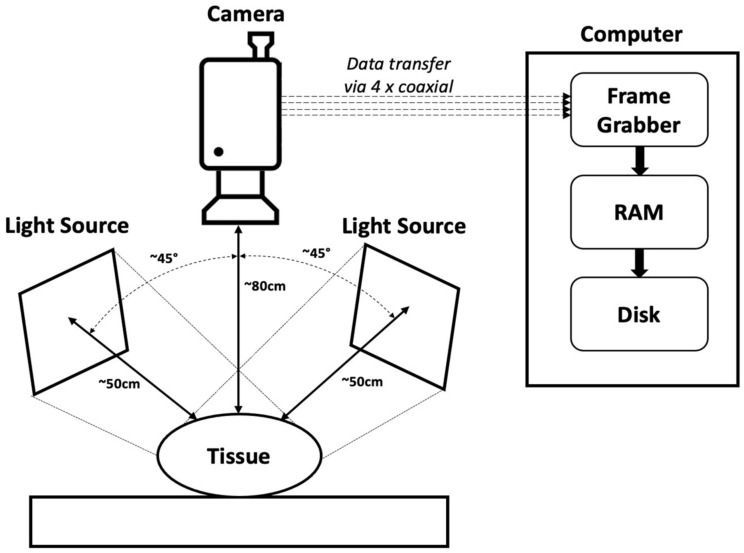
Acquisition setup, with the tissue supported by a surface and illuminated by two light sources. The camera acquires the data from above the tissue, transfers the video data via coaxial cables to the frame grabber within the acquisition computer, and then directly to the RAM. Once the video data collection is complete, the data is transferred to the disk.

**Figure 4 sensors-23-00615-f004:**
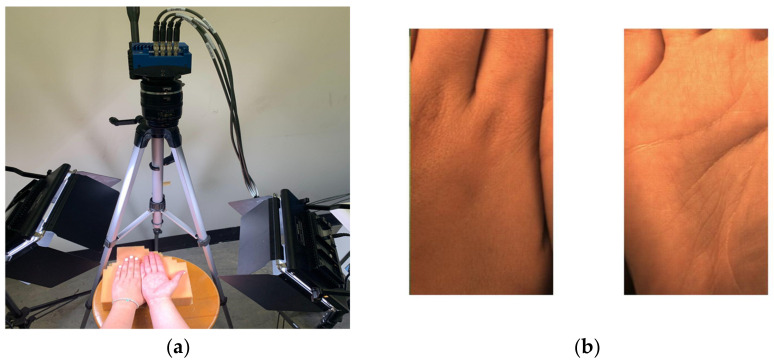
Subject undergoing signal acquisition (**a**) and subsequent division into left (non-glabrous) and right (glabrous) (**b**).

**Figure 5 sensors-23-00615-f005:**
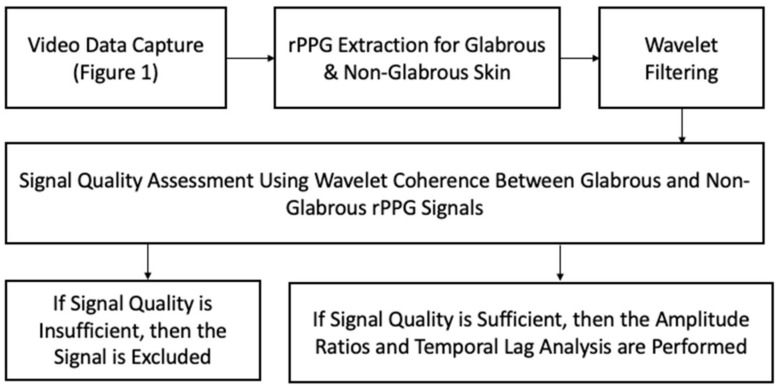
Diagram of the signal processing flow, beginning with data capture and rPPG extraction, and proceeding to filtering, signal quality assessment, and calculation of parameters.

**Figure 6 sensors-23-00615-f006:**
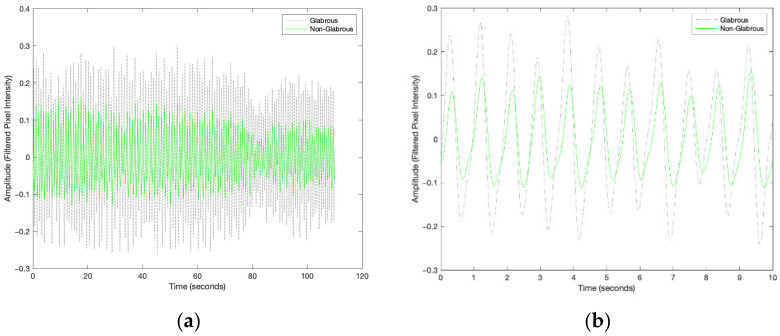
Example simultaneously acquired filtered glabrous (black) and non-glabrous (green) rPPG signals from the green channel, showing the entire duration of the acquisition (**a**) and the first 10 s (**b**).

**Figure 7 sensors-23-00615-f007:**
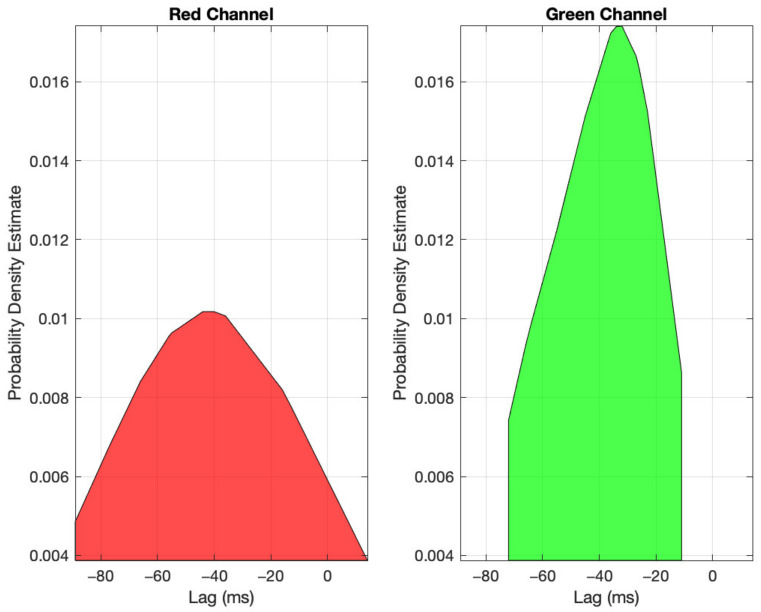
The distribution of temporal lag between glabrous and non-glabrous signals for the red and green channels, with negative lags representing glabrous leading non-glabrous.

**Table 1 sensors-23-00615-t001:** Dataset statistics, with *p*-values < 0.05 considered significant.

Parameter	Value (Mean ± SD)	*p*-Value
Amplitude Ratio in Red Channel	1.41 ± 0.85	<0.05
Amplitude Ratio in Green Channel	2.27 ± 0.88	<0.05
Lead of Glabrous over Non-Glabrous in Red Channel	32.69 ± 55.26 ms	<0.05
Lead of Glabrous over Non-Glabrous in Green Channel	40.56 ± 26.97 ms	<0.05
Lead of Red over Green in Glabrous Skin	17.13 ± 10.69 ms	<0.05
Lead of Red over Green in Non-Glabrous Skin	19.31 ± 12.66 ms	<0.05

**Table 2 sensors-23-00615-t002:** Statistically significant results for paired testing (*p* < 0.05).

Parameter	Left Hand as Glabrous Skin Source	Right Hand as Glabrous Skin Source
Amplitude Ratio in Red Channel	1.84 ± 0.81	1.09 ± 0.79
Lead of Glabrous over Non-Glabrous in Red Channel	−19.71 ± 46.96 ms	−61.29 ± 30.85 ms

## Data Availability

The data presented in this study is available upon request from the corresponding author.

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
