# Peer review of "Remote Photoplethysmography with a High-Speed Camera Reveals Temporal and Amplitude Differences between Glabrous and Non-Glabrous Skin"

_sensors, 2023, doi:10.3390/s23020615_

Round 1

Reviewer 2 Report

This paper used a high-speed scientific-grade camera to collect remote/contactless PPG from the hands (palmar/dorsal) of 22 healthy volunteers. Authors found a statistically significant difference in the amplitude of the glabrous signal over the non-glabrous signal (1.41±0.85 in the red channel and 2.27±0.88 in the green channel). In addition, they found a statistically significant lead of the red channel over the green channel, which is consistent between glabrous (17.13±10.69ms) and non-glabrous (19.31±12.66ms) skin.  I suggest that the authors use more diagrams and mathematical formulas to explain the content in detail, especially 2.2. Experimental Protocol, 2.4. rPPG Extraction and Processing.

Reviewer 3 Report

This study reports photoplethysmography of glabrous and non glabrous skin by using commercial LED light source. The overall manuscript is well written and the references are appropriate.  The nature of this work has direct applications biomedical research and possibly could be extended towards disease diagnosis. In my opinion this manuscript should be published after major revision and inclusion of necessary details as recommended below.

1.       The research should have a target oriented goal. The abstract should include the most relevant and direct application of this work in biomedical science. The last few lines in abstract should emphasis the applications. What kind of diagnosis  will requires this setup?.

2.       The last paragraph in Section-2 must explicitly highlight the novelty of this work and the improvement in results as compared to the previous studies.

3.       Section-2.1, Figure 1 should include the spectral response of the camera.

4.       Section-2.1 should present the emission spectra of the light source used in the experiment. This section should also justify that the variations in camera response and the emission from LED will not alter the outcomes of this study.

5.       Section-2.3, the subjects used in this study could have different skin thickness. Explain why it will not be a problem in this study?

6.       Figure 3 caption should be included appropriately

7.       Section-3 explain why only the Red and Green Channels of the RGB were used ?  why the Blue colour is worthless?

8.       In Section-4, Include some quantitative or qualitative description of some of its direct biomedical research application such as disease diagnosis or screening test. What details can be inferred from this study in the area of healthcare?

Reviewer 4 Report

See the PDF-file

Round 2

Reviewer 2 Report

No comments.

Author Response

Reviewer 2,

Thank you for the feedback that you have provided. We have applied an additional minor spelling and grammar check, and hope that you find it acceptable.

Reviewer 3 Report

After reading the revised manuscript and the Authors response to my comments, I agree to accept the manuscript for publication after minor grammatical corrections.

Author Response

Reviewer 3,

Thank you for the feedback that you have provided. We have applied an additional minor spelling and grammar check, and hope that you find it acceptable. 

Reviewer 4 Report

SEE THE REVIEWER COMMENTS IN PDF FILE

Author Response

Reviewer 4,

Thank you for your comments. Please find the revised text regarding the non-ideal environmental conditions and the corresponding sensitivity of the sensor follow (lines 356-360):

Further, the optimization of the acquisition setup should consider that all work to date has taken place in a laboratory environment with ideal conditions, and that data collection in future applications may take place under non-ideal conditions. Therefore, the sensitivity of the sensors to environmental conditions must be quantified and managed through design improvements of the system, as necessary